# Plant grafting relieves asymmetry of jasmonic acid response induced by wounding between scion and rootstock in tomato hypocotyl

Jiaqi Wang[1,2,3,4,5], Dongliang Li[3,4], Ni Chen[2,3,4], Jingjing Chen[3,4], Changjun Mu[6], Kuide Yin[1]*, Yuke He[5]*, Heng Liu [1,2,3,4,5]*

1 Plant and Microbe Interaction Lab, Hei Longjiang Bayi Agricultural University, Daqing, Hei Longjiang, P. R. China, 2 College of Life Science, Shaoxing University, Zhejiang, P. R. China, 3 South Subtropical Crop Research Institute, Chinese Academy of Tropical Agricultural Sciences, Ministry of Agriculture, Zhanjiang, Guangdong, P. R. China, 4 National Field Genebank for Tropical Fruit, Institute of South Subtropical Crop Research Institute, Chinese Academy of Tropical Agricultural Science, Zhanjiang, China, 5 National Key Laboratory of Plant Molecular Genetics, Shanghai Institute of Plant Physiology and Ecology, Chinese Academy of Sciences, Shanghai, P. R. China, 6 Ministry of Education Key Laboratory of Cell Activities and Stress Adaptations, School of Life Sciences, Lanzhou University, Lanzhou, China

* yinkuide@163.com (KY); yuke@sibs.ac.cn (YH); hengliu@vip.163.com (HL)

**Data Availability Statement:** All relevant data are within the manuscript and its Supporting Information files.

**Funding:** This work was supported by grants from Central Public-interest Scientific Institution Basal

## Abstract

Plant grafting is a sequential wound healing process. However, whether wounding induces a different jasmonic acid (JA) response within half a day (12 h) after grafting or non-grafting remains unclear. Using the tomato hypocotyl grafting method, we show that grafting alleviates the asymmetrical accumulation of JA and jasmonic acid isoleucine conjugate (JA-Ile) in scion and rootstock caused by wounding, and from 2 h after tomato micrografting, grafting obviously restored the level of JA-Ile in the scion and rootstock. Meanwhile, five JA-related genes, *SlLOX11*, *SlAOS*, *SlCOI1*, *SlLAPA* and *SlJA2L*, are detected and show significant changes in transcriptional expression patterns within 12 h of grafting, from asymmetrical to symmetrical, when the expression of 30 JA- and defense-related genes were analyzed. The results indicated that grafting alleviates the asymmetrical JA and defense response between scion and rootstock of the tomato hypocotyl within 12 h as induced by wounding. Moreover, we demonstrate that in the very early hours after grafting, JA-related genes may be involved in a molecular mechanism that changes asymmetrical expression as induced by wounding between scion and rootstock, thereby promoting wound healing and grafting success.

## Introduction

Plant grafting is an asexual propagation technique that is used to improve disease resistance, control vigor, increase yield and fruit quality, regulate precocity and plant size, and adapt to biotic and abiotic stress in horticulture [1]. For example, in tomatoes (*Solanum lycopersicum*), to control soilborne pathogens, grafting is widely used as an alternative to the use of methyl bromide [2]. Successful grafting relies on the accurate recognition and crosstalk between rootstock and scion [3]. Recently, a model has been proposed to illustrate the stages of graft union

Research Fund for the Chinese Academy of
Tropical Agricultural Sciences (No.
1630062020001), and Species and Varieties
Resources Protection Project of Ministry of
Agricultural and Rural Affairs
(No.125163006000160004). The funders had no
role in study design, data collection and analysis,
decision to publish, or preparation of the
manuscript.

**Competing interests:** The authors have declared
that no competing interests exist.

formation, and phytohormones may be involved and regulate each stage of this process [4–6].
The role of auxin in vascular system reconstruction during graft has been gradually revealed
[4, 7–9]. However, little is known about the roles of jasmonic acid (JA) in grafting, although
some transcriptome data show that grafting induces the expression of JA biosynthesis and sig-
naling genes on the first day of Arabidopsis hypocotyl grafting and watermelon heterograft
combinations [4, 10]. Interestingly, Arabidopsis *AOS* mutants with disrupted JA biosynthesis
could be grafted successfully [11].

Wounding response is the first issue when graft or wounding disconnects plant tissues. In
Arabidopsis inflorescence stems incised to half-diameter depth, the plant hormone auxin pro-
motes the division of pith cells and wound healing, and ethylene and JA are also involved in
the wound healing response [12–14]. Around the cut site, two transcriptional factor (TF)
genes were selected by microarray analysis. *ANAC071* was expressed exclusively at the top of
the cut gap, whereas *RAP2.6L* was expressed after cutting exclusively at the bottom [12]. At the
graft junction of Arabidopsis hypocotyls, tissues initially show asymmetry in cell division, cell
differentiation, and gene expression, when the scion and rootstock tissue come into contact;
this type of asymmetry will be relieved and the vascular connection start to form [8]. Two
genes involved in auxin perception, namely, *ALF4* and *AXR1*, play more important roles in the
bottom but not the top of the graft junction for phloem connection [8]. When tissues around
the junction were isolated completely at different time points (6 h to 240 h) after grafting and
using cut tissues that remained separated as control, the majority of genes were expressed
asymmetrically at the scion and rootstocks. As the graft temporally healed, this asymmetry
gradually disappeared, and the vasculature reconnected [15]. However, wounding induces a
JA response, and the accompanying wounding- associated asymmetry processes involved in
the early hours (before 6 h) of grafting remain unclear.

The present study investigated how wounding induces a JA response and its related gene
expression patterns during 12 h of graft formation using a well-established tomato hypocotyl
grafting method [4]. By measuring the content of JA and jasmonic acid isoleucine conjugate
(JA-Ile) at 12 h after grafting, we reveal that grafting alleviates the asymmetrical accumulation
of JA and JA-Ile in the scion and rootstock caused by wounding, and from 2 h after grafting,
grafting obviously restored the level of JA-Ile in the scion and rootstock. Meanwhile, the
expression patterns of 30 JA- and defense-related genes within 12 h after grafting were ana-
lyzed, five genes showed significant changes in expression patterns after grafting, from asym-
metrical to symmetrical, suggesting that grafting alleviates the asymmetrical JA and defense
response between scion and rootstock induced by wounding within 12 h. We propose that in
the early hours after grafting, JA-related genes act as part of a mechanism that changes asym-
metry that is induced by wounding, thereby promoting wound healing and grafting success.

## Results

### Dynamic behavior of JA and JA-Ile accumulation within 12 h after grafting

To understand the dynamic accumulation of JAs at the first few hours after grafting, hypocotyl
grafting of tomato was performed and the levels of JA and JA-Ile were measured at six time
points (0, 0.5, 1, 2, 6, and 12 h) after grafting (Fig 1A). Scions that were cut off and not grafted
(separated top) were chosen as the control of grafted scion. Similarly, rootstocks that had been
cut off and not grafted (separated bottom) were selected as the control of grafted rootstock.
The control group was sampled at the same six time points as the grafting group. When we
analyzed the hormone levels of grafted scion and rootstock (or the top and bottom of the sepa-
ration), a smaller difference revealed a lower degree of asymmetry, and vice versa. Our results
show a significant difference in the accumulation of JAs between the top and bottom

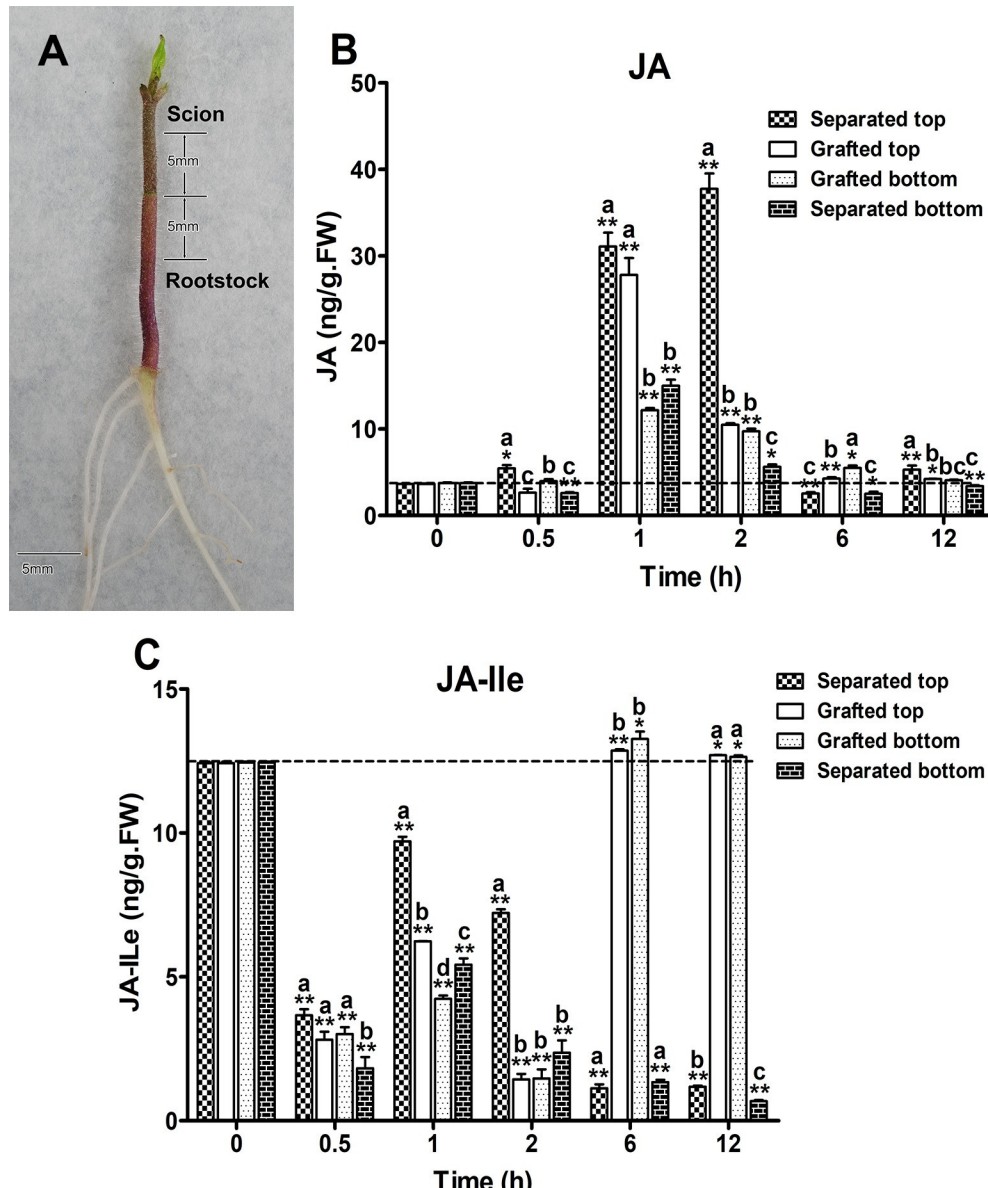

**Fig 1. Jasmonic acid (JA) and jasmonic acid isoleucine conjugate (JA-Ile) dynamical content in tomato within 12 h of grafting.** (A) Tomato hypocotyl 12 h after grafting. The top part is the scion, and the bottom part is the rootstock. The sampling distance is 5 mm above and below the joint site. Bar = 5 mm. (B) JA content changed dynamically within 12 h of grafting. (C) JA-Ile content changed dynamically within 12 h of grafting. (B-C) Coordinate axis: X axis is the time point after graft or separated; Y axis is the hormone level. Separated: hypocotyl cut off without grafting; Grafted: grafting after hypocotyl cutting; Bottom: rootstock that was grafted or bottom not grafted; Top: scion that was grafted or top not grafted. Values are expressed as the mean ± SE of three independent. Each time points compared with 0 h control, $^{*}P \leq 0.05$ and $^{**}P \leq 0.01$ in one-tail Student's $t$-test (no mark was $P > 0.05$, the difference was not significant). Comparison between each group; Those marked with different letters were $P \leq 0.05$, has a significant difference in one-tail Student's $t$-test (Contains the same letter as $P > 0.05$, the difference is not significant). Dotted line: hormone content at 0 h.

(separated) or the scion and rootstock (grafted) parts within first 12 h (Fig 1B; see also S1 Table). For the separated top, JA levels were significantly higher than the grafted scion from 0.5 h, about 2-fold of that observed in the scion, reached the second peak in 1 h, continued to rise for 2 h to reach the peak, and then sharply decreased, and reached the lowest at 6 h (Fig

1B). The scion, however, peaked at 1 h, followed by a rapid decline. At 2 h, JA content in the scion was only about 1/4 of that in the separated top, followed by a rapid decline until 6 h later, and then slowed down to the level of 0 h (Fig 1B). For the rootstocks, compared to the sharply decrease in the separated bottom, the overall decreasing pattern was delayed. The decrease rate of the grafting was nearly 2/3 compared to that without grafting (Fig 1B). After grafting, the JA levels of the scion and rootstock after 2 h tend to be similar, means less asymmetrical, but the separated top and bottom took 6 h to converge, means more asymmetrical. Our results indicate that grafting reduces the difference in JA content between the top part of the cut plant (scion) and the bottom part (rootstock), resulting in a synchronous or symmetrical response.

Furthermore, we determined the content of JA-Ile, the active substance of JA, within first 12 h after grafting (Fig 1C; see also S1 Table). For the graft, both the scion and rootstock showed a "W"-shaped response (decreasing/increasing/decreasing/increasing) in 0–6 h (Fig 1C). The bottom of the wave appeared at 0.5 and 2 h. The response of the bottom within 0–2 h was the same as that of the scion, but there was no significant increase at 6 h, and a continuous, slow decline was observed up to 12 h. For the top, the trend was similar to that of the bottom, and the inflection point was the same, but the JA-Ile levels at 0.5, 1, 2, and 6 h were significantly higher than the bottom at their respective time points, until 12 h similar with the bottom (Fig 1C). Therefore, in the case of separated parts, the JA-Ile response pattern was similar in the top and bottom, but the top was significantly higher than the bottom at 4 time points (0.5–6 h), indicating that the separated top had a more obvious response. In the case of grafting, the response of JA-Ile in the scion and rootstock was similar. After grafting, the content of JA-Ile in the grafted seedlings returned to the normal level at 6 h, but the level of JA-Ile at the top and bottom without grafting could not be recovered until 12 h.

## Expression of six JA synthesis-related genes 12 h after grafting

We further studied the expression patterns of 30 JA-related genes within 12 h of grafting. First, six JA synthesis-related genes were analyzed, which included SOLANUM LYCOPERSICUM TOMATO LIPOXYGENASE D (*SlTomLoxD*), *LOX11*, SOLANUM LYCOPERSICUM ALLENE OXIDE SYNTHASE (*SlAOS*), SOLANUM LYCOPERSICUM ALLENE OXIDE CYCLASE(*SlAOC*), SOLANUM LYCOPERSICUM OXOPHYTODIENOATE-REDUCTASE3 (*SlOPR3*), and SOLANUM LYCOPERSICUM opc-8:0 CoA Ligase1 (*SlOPLC1*) (Fig 2; see also S2 Table and S1 Fig).

The results show that in both scion and rootstock, these synthetic genes showed a significant peak of expression from 0.5 to 2 h after grafting, and the change was significant compared with that of no grafting at those time points. Furthermore, the *SlTomLoxD* response was the earliest. The peak value in the scion after grafting was maintained until 1 h before it began to decrease. A short peak plateau period was formed between 0.5 and 1 h, but the rootstock did not have this plateau period. The expression of the *SlAOC* gene in the scion forms a short peak plateau period between 1 h and 2 h, but in the rootstock, *SlAOS* and *SlAOC* did not form an obvious plateau period during this period. For *SlOPR3* and *SlOPLC1*, neither scion nor rootstock showed any plateau period. In the separated tissues, *SlAOS*, *SlAOC*, *SlOPR3*, and *SlOPLC1* all failed to form significant peaks at the top and bottom, except for the obvious peak of *SlTomLoxD*. Interestingly, the expression of *SlLOX11* showed significantly different trends within 12 h. After grafting, *SlLOX11* peaked in the scions and rootstocks at 6 h and then decreased. However, in the top of the separated, there was a short rise and fall at 0.5 h after the cut off, and then continued to increase. The role of *SlLOX11* in injury and grafting requires further investigation. In general, for the peak expression level, the grafted group was higher than the separated, except for *SlTomLoxD* and *SlLOX11* in the top, which was significantly

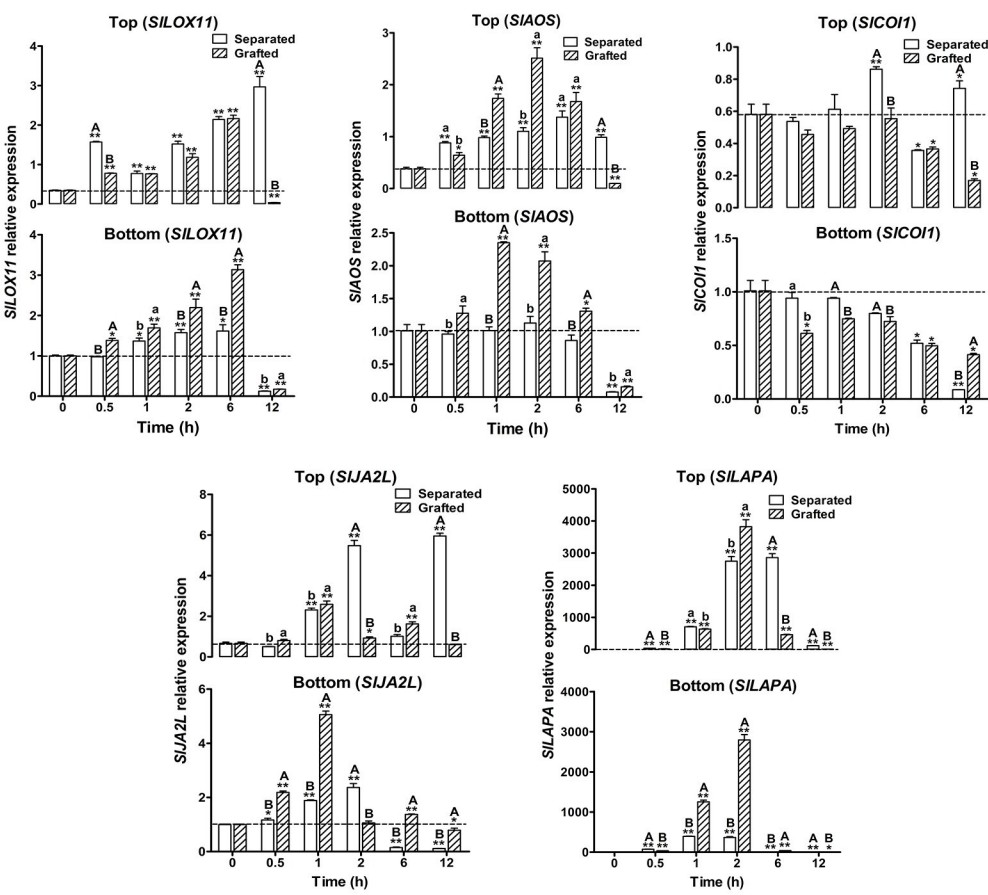

**Fig 2. Dynamic changes in the expression of JA -related genes within 12 h of grafting.** The separated part was used as control. Values are expressed as the mean ± SE of three independent replicates. Coordinate axis: X axis is the time point after graft or separated; Y axis is the gene relative expression level. Separated: hypocotyl cut off without grafting; Grafted: grafting after hypocotyl cutting; Bottom: rootstock that was grafted or bottom not grafted; Top: scion that was grafted or top not grafted. Each time points compared with 0 h control, $^{*}P \leq 0.05$ and $^{**}P \leq 0.01$ in one-tail Student's $t$-test (no mark was $P > 0.05$, the difference was not significant). Grafted part compared with separated part control; "ab" $P \leq 0.05$; "AB" $P \leq 0.01$ in one-tail Student's $t$-test (no mark was $P>0.05$, the difference is not significant). Dotted line: gene relative expression level at 0 h.

higher than the grafted. The above results indicate that grafting significantly increases the expression of JA synthesis-related genes in the scion and rootstock and play an important role in the early stage of grafting.

## Expression of seven JA signaling genes within 12 h of grafting

Similar to the expression pattern analysis of JA synthesis-related genes, we analyzed seven genes associated with the JA signaling pathway within 12 h of grafting, including SOLANUM LYCOPERSICUM CORONATINE INSENSITIVE 1 (*SlCOI1*), SOLANUM LYCOPERSICUM MYELOCYTOMATOSIS PROTINS 2 (*SlMYC2*), SOLANUM LYCOPERSICUM JASMONA-TE-ZIM DOMAIN 1 (*SlJAZ1*), *SlJAZ2*, *SlJAZ3*, *SlJAZ5*, and *SlJAZ6* (Fig 2; see also S2 Table and S2 Fig).

The expression of *SlCOI1* gradually decreased in the scion and rootstock within 12 h of grafting. The downregulation of *SlCOI1* in the separated bottom was similar to the rootstocks, and the expression curve of the bottom without grafting was relatively steep. Unlike the

gradual decrease in the scion after grafting, *SlCOI1* expression fluctuated in the scion without grafting within 12 h, suggesting that grafting altered the expression of *SlCOI1*. However, *SlMYC2*, *SlJAZ1*, *SlJAZ2*, *SlJAZ3*, *SlJAZ5*, and *SlJAZ6* showed significant peaks in expression in the top and bottom sections, regardless of grafting. As a hub gene in pathways such as injury and disease resistance, grafting significantly increased the expression of *SlMYC2* in the scion and rootstock, both of which rapidly increased within 1 h and then immediately decreased. The peak expression of *SlJAZ1*, *SlJAZ2*, and *SlJAZ3*, after grafting in the scion and rootstock was delayed by 30 min, but *SlJAZ5* and *SlJAZ6* showed no obvious delay or advancement. Among the five *SlJAZ* genes, the expression level of *SlJAZ6* after grafting was lower than that of the separated region, especially in the scions, while the expression level of the other *SlJAZ* genes significantly increased in the scions and rootstocks after grafting.

## Expression of six defense genes within 12 h of grafting

Six defense genes were analyzed within 12 h of grafting, including SOLANUM LYCOPERSI-CUM THREONINE DEAMINASE (*SlTD*), SOLANUM LYCOPERSICUM LEUCINE AMI-NOPEPTIDASE A (*SlLAPA*), SOLANUM LYCOPERSICUM PROTEINASE INHIBITOR-1 (*SlPI-1*), and SOLANUM LYCOPERSICUM PATHOGENESIS RELATED STH-2 (*SlPR-STH2*), SOLANUM LYCOPERSICUM ARGINASE (*SlARG*), and SOLANUM LYCO-PERSICUM JA2-LIKE (*SlJA2L*) (Fig 2; see also S2 Table and S3 Fig).

In the scions, grafting significantly increased the expression of *SlPI-1*, although the increasing trend within 12 h was similar with the separated part, both reaching a peak at 6 h. In root-stocks, grafting significantly increased expression in the rootstock and continued to increase within 2 h, peaking at 2 h, and then decreased. However, in the separated group, expression peaked at 0.5 h and then dropped. This indicates that the response of *SlPI-1* is clearly different in the top and bottom without grafting. After grafting, they tend to be consistent and more responsive in the rootstock. In tomato, *SlTD* is commonly used as a marker gene for wounding or insect response and *SlPR-STH2* for pathogen response [16]. In the top part, the *SlTD* gene peaked at 6 h with or without grafting, but the expression significantly decreased after grafting.

The expression of *SlPR-STH2* in the top and bottom, whether grafted or not, peaked at 1 h and then decreased. Moreover, after grafting, the expression of *SlPR-STH2* in the scion and rootstock is significantly higher than that of the separated. The *SlARG* expression profile in the top with or without grafting was consistent. In the bottom, the separated group peaked at 2 h and the grafted group at 6 h. This indicates that the response time of the rootstock and scion is consistent after grafting. Without grafting, *SlLAPA* expression peaked at 2 to 6 h in the top and 1 to 2 h in the bottom. After grafting, *SlLAPA* expression in the scion and rootstock reached a distinct peak at 2 h, indicating that the expression of *SlLAPA* in the scion and rootstock tended to be temporally consistent after grafting. Meanwhile, the results show that the expression of *SlLAPA* after grafting was significantly higher than that of the separated. In the separated top, the expression of *SlJA2L* peaked at 2 h, decreased at 6 h, and reached the same peak level at 12 h, clearly showing fluctuations. After grafting, expression peaked at 1 h, followed by fluctuations, but the amplitude was significantly lower than that of the separated, and by 12 h, expression significantly decreased. In the bottom, the expression of *SlJA2L*, was similar to that in the top and peaked at 2 h. After grafting, it also peaked at 1 hour, similar to that in the scion, indicating that grafting makes the expression in the scion and rootstock similar.

## Expression of 11 JA-regulated genes within 12 h of grafting

We finally analyzed 11 genes that possibly regulate the JA signaling pathway within 12 h of grafting, including Solyc06g075510 SOLANUM LYCOPERSICUM ETHYLENE RESPONSE

FACTOR (*SlERF*), Solyc01g090320 (*SlERF*), Solyc01g090560 (*SlERF*), and Solyc02g070040 (*SlERF*), BASIC HELIX-LOOP-HELIX (*SlbHLH*), Solyc09g083360 *SlbHLH*, Solyc05g050560 *SlbHLH*, SOLANUM LYCOPERSICUM HOMEODOMAIN-LEUCINE-ZIPPER (*SlHD-Zip*), *SlTCP 5*, SOLANUM LYCOPERSICUM MITOGEN-ATIVATED PROTEIN KINASES 1 (*SlMPK1*), *SlMPK2*, and *SlMPK3*, and 8 TF genes were included, all of which were *MYC2*-targeted TF genes that responded to wounding and was mentioned in the Du 2017 report (Fig 2; see also S2 Table and S4 Fig) [16].

For the four *SlERF*s, except for the expression of the Solyc01g09032*ERF* gene in the scion, the other three *SlERF*s showed the same trend, i.e., enhanced after grafting, and the peak was delayed by 30 min and reached the peak at 1 h. Grafting had no significant effect on the expression of Solyc09g083360*bHLH* in scions, while grafting decreased the expression of Solyc09g083360*bHLH* at 0.5 h but increased expression at 1 h in the rootstock. For *SlHD-Zip*, grafting significantly increased its expression in scions and rootstocks at 1 h. *SlTCP5* transcription factors were also induced in grafted scions and rootstocks. The expression of three *SlMPK* genes within 12 h of grafting exhibited their own characteristics. Irrespective of grafting, *SlMPK3* expression peaked at 0.5 h. Grafting reduced the peak *SlMPK3* expression levels in the scions at 0.5 h, but increased the expression in the rootstock at 0.5 h. Grafting increased the expression of *SlMPK1* and *SlMPK2* in the scions and rootstock, and there was no obvious peak when grafting was not performed. After grafting, *SlMPK1* showed low peak expression in scions for 6 h and rootstock for 2 h, and *SlMPK2* formed a low flat peak in the rootstock at 1 h, whereas no significant peak was observed in the scions.

## Effects of grafting on the asymmetry of JA-related genes expression patterns

The issue of asymmetric gene expression during grafting deserves further study [15]. In this study, we used the correlation coefficient *r* and the determination coefficient $r^2$ to represent the symmetry of the top and bottom parts of the graft. The smaller the determination coefficient, the worse the symmetry of gene expression between the top and bottom parts, and the higher the asymmetry. The greater the determination coefficient, the closer the expression of genes of the top and bottom parts of the grafting junction, and the better the symmetry. Our results showed that in the separated parts, 2 of the 6 genes related to JA synthesis were asymmetric (Table 1), 2 of the 7 genes related to JA signaling were asymmetric (Table 2), 5 of the 6 genes related to defense were asymmetric (Table 3), and 3 of the 11 genes related to regulation were asymmetric (Table 4). Overall, there were 12 genes with asymmetric expression patterns in the junction without grafting. After grafting, the results showed that all the six genes related to JA synthesis showed symmetry (Table 1). Six of the seven JA signaling genes showed symmetry (Table 2). Three of the six genes for defense expressed symmetry (Table 3). Among the 11 regulated genes, 7 showed symmetry (Table 4). In general, within 12 h of grafting, there were 23 genes whose expression patterns were symmetrical and 7 genes whose expression patterns were asymmetric. After grafting, the asymmetrically expressed genes were mainly those related to defense and signal regulation. Under the influence of grafting, the expression of five genes changed from asymmetrical to symmetrical. These five genes are *SlLOX11*, *SlAOS*, *SlCOI1*, *SlJA2L*, and *SlLAPA*.

## Discussion

### Grafting alleviates the asymmetrical accumulation of JA in scions and rootstocks caused by wounding

Plant grafting is a complex wound healing and tissue regeneration process that is dependent on local signal communication between scion and rootstock genotypes [17]. Wound signals

**Table 1. Symmetry of the expression patterns of JA synthesis-related genes in the top (scion) and bottom (rootstock) of tomato hypocotyls within 12 h of grafting.**

| Genes | Groups | Pearson correlation ($r$) | $P$ value | $r^2$ |
|---|---|---|---|---|
| SlAOS | Separated | -0.119 | 0.822 | 0.01427 |
|  | Grafted | 0.841 | 0.036 | 0.7074 |
| SlAOC | Separated | 0.934 | 0.006 | 0.8724 |
|  | Grafted | 0.98 | 0.001 | 0.9606 |
| SlLOX11 | Separated | -0.454 | 0.366 | 0.2062 |
|  | Grafted | 0.976 | 0.001 | 0.9534 |
| SlTomLoxD | Separated | 0.995 | 0.0007 | 0.9892 |
|  | Grafted | 0.93 | 0.007 | 0.8655 |
| SlOPR3 | Separated | 0.97 | 0.001 | 0.9414 |
|  | Grafted | 0.998 | 0.0005 | 0.9954 |
| SlOPCL1 | Separated | 0.978 | 0.001 | 0.9564 |
|  | Grafted | 0.995 | 0.0007 | 0.9891 |

Comparison between the top (scion) and bottom (rootstock): $P > 0.05$, no correlation; $P \leq 0.05$, significant correlation; and $P \leq 0.01$, extremely significant correlation. Pearson correlation coefficient, $0.8 \leq |r| \leq 1$, highly correlated; $0.5 \leq |r| < 0.8$, moderately correlated; $0.3 \leq |r| < 0.5$, weakly correlated, and $|r| < 0.3$, negligible correlated. $r^2$ is the determining coefficient.

are perceived through so-called damage-associated molecular patterns (DAMP), and then JA synthesis is upregulated and defense responses are activated [18]. JA accumulated in Arabidopsis leaves within 30 s of injury [19, 20]. In Arabidopsis leaves that were completely cut off, JA remained at the peak level from 10 min to 1 h, then decreased, and the content of JA-Ile peaked at about 30 min, and then began to decline. Both JA and JA-Ile could recover levels observed at 0 h at 4 h [21]. After wounding, peak expression was observed at 1 h in tomato leaves [22]. In Arabidopsis roots, after trauma, the accumulation of JA and JA-Ile was similar to that in leaves, and root synthesis of JA and JA-Ile was independent of the shoots and leaves

**Table 2. Symmetry of the expression patterns of JA signaling genes in the top (scion) and bottom (rootstock) of tomato hypocotyls within 12 h of grafting.**

| Genes | Groups | Pearson correlation ($r$) | $P$ value | $r^2$ |
|---|---|---|---|---|
| SlJAZ1 | Separated | 0.856 | 0.03 | 0.7331 |
|  | Grafted | 0.937 | 0.006 | 0.8779 |
| SlJAZ2 | Separated | 0.993 | 0.0007 | 0.9863 |
|  | Grafted | 0.918 | 0.01 | 0.8425 |
| SlJAZ3 | Separated | 0.964 | 0.002 | 0.9302 |
|  | Grafted | 0.989 | 0.0008 | 0.9775 |
| SlJAZ5 | Separated | 0.966 | 0.002 | 0.9336 |
|  | Grafted | 0.971 | 0.001 | 0.943 |
| SlJAZ6 | Separated | 0.96 | 0.002 | 0.9209 |
|  | Grafted | 0.925 | 0.008 | 0.8559 |
| SlCOI1 | Separated | -0.123 | 0.817 | 0.01508 |
|  | Grafted | 0.872 | 0.024 | 0.7604 |
| SlMYC2 | Separated | 0.6 | 0.208 | 0.36 |
|  | Grafted | 0.751 | 0.085 | 0.5647 |

Comparison between the top (scion) and bottom (rootstock): $P > 0.05$, no correlation; $P \leq 0.05$, significant correlation; and $P \leq 0.01$, extremely significant correlation. Pearson correlation coefficient, $0.8 \leq |r| \leq 1$, highly correlated; $0.5 \leq |r| < 0.8$, moderately correlated; $0.3 \leq |r| < 0.5$, weakly correlated, and $|r| < 0.3$, negligible correlated. $r^2$ is the determining coefficient.

**Table 3. Symmetry of the expression patterns of defense genes in the top (scion) and bottom (rootstock) of tomato hypocotyls within 12 h of grafting.**

| Genes | Groups | Pearson correlation ($r$) | $P$ value | $r^2$ |
|---|---|---|---|---|
| SlPI-1 | Separated | -0.755 | 0.083 | 0.5693 |
| | Grafted | 0.11 | 0.835 | 0.0122 |
| SlJA2L | Separated | 0.148 | 0.779 | 0.02196 |
| | Grafted | 0.87 | 0.024 | 0.7569 |
| SlTD | Separated | 0.271 | 0.68 | 0.04702 |
| | Grafted | 0.587 | 0.22 | 0.3451 |
| SlLAPA | Separated | 0.313 | 0.546 | 0.09782 |
| | Grafted | 0.947 | 0.004 | 0.897 |
| SlPR-STH2 | Separated | 0.954 | 0.003 | 0.9107 |
| | Grafted | 0.977 | 0.001 | 0.9539 |
| SlARG | Separated | -0.107 | 0.84 | 0.01144 |
| | Grafted | 0.569 | 0.238 | 0.324 |

Comparison between the top (scion) and bottom (rootstock): $P > 0.05$, no correlation; $P \leq 0.05$, significant correlation; and $P \leq 0.01$, extremely significant correlation. Pearson correlation coefficient, $0.8 \leq |r| \leq 1$, highly correlated; $0.5 \leq |r| < 0.8$, moderately correlated; $0.3 \leq |r| < 0.5$, weakly correlated, and $|r| < 0.3$, negligible correlated. $r^2$ is the determining coefficient.

**Table 4. Symmetry of the expression patterns of JA signal regulated genes in the top (scion) and bottom (rootstock) of tomato hypocotyl after grafting within 12 h.**

| Genes | Groups | Pearson correlation ($r$) | $P$ value | $r^2$ |
|---|---|---|---|---|
| SlERF (Solyc01g090320) | Separated | 0.855 | 0.03 | 0.7308 |
| | Grafted | 0.567 | 0.24 | 0.322 |
| SlERF (Solyc01g090560) | Separated | 0.882 | 0.02 | 0.7783 |
| | Grafted | 0.994 | 0.0007 | 0.9875 |
| SlERF (Solyc02g070040) | Separated | 0.908 | 0.012 | 0.8253 |
| | Grafted | 0.936 | 0.006 | 0.8762 |
| SlTCP5 | Separated | -0.238 | 0.65 | 0.05652 |
| | Grafted | 0.792 | 0.061 | 0.6268 |
| SlERF (Solyc06g075510) | Separated | 0.838 | 0.037 | 0.7017 |
| | Grafted | 0.976 | 0.001 | 0.9527 |
| SlHD-Zip | Separated | 0.893 | 0.016 | 0.798 |
| | Grafted | 0.926 | 0.008 | 0.8582 |
| SlbHLH (Solyc09g083360) | Separated | 0.887 | 0.018 | 0.7868 |
| | Grafted | 0.907 | 0.013 | 0.8231 |
| SlbHLH (Solyc05g050560) | Separated | 0.955 | 0.003 | 0.9118 |
| | Grafted | 0.965 | 0.002 | 0.9313 |
| SlMPK1 | Separated | -0.629 | 0.181 | 0.3953 |
| | Grafted | 0.73 | 0.1 | 0.5326 |
| SlMPK2 | Separated | 0.062 | 0.906 | 0.0039 |
| | Grafted | 0.791 | 0.061 | 0.6254 |
| SlMPK3 | Separated | 0.997 | 0.0006 | 0.9935 |
| | Grafted | 0.999 | 0.0003 | 0.9989 |

Comparison between the scion and rootstock: $P > 0.05$, no correlation; $P \leq 0.05$, significant correlation; and $P \leq 0.01$, extremely significant correlation. Pearson correlation coefficient, $0.8 \leq |r| \leq 1$, highly correlated; $0.5 \leq |r| < 0.8$, moderately correlated; $0.3 \leq |r| < 0.5$, weakly correlated, and $|r| < 0.3$, negligible correlated. $r^2$ is the determining coefficient.

[23]. Our study showed that in the case of separated tomato hypocotyls, transient JA accumulated in the top and bottom parts. Peak expression in the top and bottom parts were observed at 1 h, but the top part peak lasted for 2 h and maintaining a significant 1-hour peak platform, then dropped. However, the bottom peak immediately decreased after 1 h. After grafting, the 1-h peak platform in the scion disappeared. In addition, within 12 h, the content of JA in grafted scions was significantly lower than that in the top part of the separated portion. This indicated that grafting could significantly alleviate JA synthesis in scion caused by wounding. Meanwhile, after the separation of the top and bottom parts of the tomato hypocotyl, the JA content in the top and bottom parts of the hypocotyl showed asymmetrical expression within 12 h, but was alleviated by grafting. Similar results were found in terms of the JA-Ile contents of scions and rootstocks before and after tomato hypocotyl grafting within 12 h. Unlike the time response curve of JA, our most interesting results was that JA-Ile content showed a "W"-shaped response in 0–6 h. Meanwhile, grafting obviously initiated to restore the level of JA-Ile in the scions and rootstocks from 2 h after grafting. In leaves, roots and shoots, generally, the JA and JA-Ile content are consistently induced by the wounding [21–23], and no JA-Ile reduction was reported within half an hour after injury in these tissues yet. We do not know whether the W-shaped JA-Ile response induced by wounding is specific to the hypocotyl of tomatoes. Therefore, we need to further study to confirm the ubiquity of this result in stem grafting of tomatoes and other species including *Arabidopsis thaliana*. Moreover, further analysis on various JA synthesis mutants will certainly contribute to our understanding of this issue.

## Grafting alleviates the asymmetrical JA response between scions and rootstocks induced by wounding within 12 h

Transcriptome analysis within 10 days after hypocotyl grafting in Arabidopsis revealed that many signaling pathways genes, especially sugar, show asymmetrical in scions and rootstocks, whose mutual attachment is a key step in the recognition and activation of graft union formation development [15]. Our further hypothesis is that the difference in physiological response between the rootstock and scion is due to variations in gene expression between the scion and rootstock after grafting, as well as the incompatibility of grafting and the failure of grafting, which can be defined as the asymmetry of grafting (AG). No difference in gene expression was observed between scions and rootstocks after grafting, indicating similar physiological responses and affinity for grafting, ultimately leading to the success of grafting, which can be defined as the symmetry of grafting (SG). Therefore, it can be considered that AG is the molecular basis of graft incompatibility, while SG is the molecular basis of graft compatibility. Thus, if the scion and rootstock are cut off and not grafted because of their respective differences in response to wounding, then the expression of certain pathway genes between the scion and rootstock is asymmetrical. Our results are similar to those of Melnyk's [15]. If the scion and stock are grafted immediately after these are cut off, then two situations may develop. First, for self-grafting, homografts, or graft compatibility, the asymmetry caused by wounding may be gradually relieved, and the asymmetry will become symmetrical, so that grafting ultimately succeeds. Second, if the rootstock and scion come from different genotypes or heterografts or if the graft is incompatible, then asymmetry may persist and hardly be converted from asymmetrical to symmetrical, resulting in graft incompatibility and graft failure. This means that it is possible to reveal the molecular basis of graft incompatibility by looking for genes that exhibit significant changes in symmetry before and after grafting. Among the 30 genes tested in our research, 5 genes showed significant changes in expression patterns after grafting, from asymmetrical to symmetrical, including synthetic genes *SlLOX11* and *SlAOS*, signaling genes *SlCOI1*, and defense genes *SlLAPA* and *SlJA2L*.

*LOX* and *AOS* plays a key role in the biosynthesis of JA. Wounding induces the accumulation of *AtLOX2* and *AtLOX6* transcripts, which are associated with long-distance signaling in Arabidopsis [19, 24]. The *AtLOX6* defective mutant could significantly reduce the accumulation of JA and JA-Ile within 190 s of leaf wounding, indicating that *AtLOX6* is clearly expressed at the very early stage after wounding. The homologous gene of Arabidopsis *AtLOX6* in tomato is *SlLOX11*, and during the early stage of mechanical damage of tomato leaves, its expression was upregulated 3–6 h after wounding [25]. Our results show that after grafting, *SlLOX11* expression peaked at 6 h in the both scions and rootstocks, and then decreased, which was similar to the results in tomato damaged leaves, but the peak time was later than that using ungrafting. Compared with other JA synthesis genes, the expression pattern of *SlLOX11* was significantly different within 12 h of grafting, changing from asymmetrical to symmetrical, suggesting that the role of *SlLOX11* in grafting development deserves further investigations. Many JA biosynthesis-related genes are still JA-inducible and are involved as *AOS*s [26]. Upon potato tuber injury, *StAOS2* is immediately upregulated from an almost undetectable level at 0 h, reaching a peak at 2 h, and decreasing to less than half of the maximum expression within 6 h after wounding [27]. In the leaves of wild-type Arabidopsis, the expression level of *AtAOS* peaked at 2 h after wounding, and then gradually decreased to normal levels [28]. Through a $Ca^{2+}$/calmodulin-dependent mechanism, *JAV1* is phosphorylated after wounding, and this leads to the destruction of *JAV1*, which finally induces *AOS* expression [29]. After tomato leaf damage, *SlAOS* begins to accumulate after 1 h, peak after 4 to 8 h, and then decline [30]. In our experiment, the grafted site was the hypocotyl. After the hypocotyl was cut off and separated, *SlAOS* expression peaked at 6 h, the bottom *SlAOS* peaked at 2 h, and then continued to decline. After grafting, both rootstock and scion *SlAOS* reached its highest expression within 2 h. Although Arabidopsis *SlAOS* mutants can be grafted successfully [11], our results still show that the expression patterns of *SlLOX11* and *SlAOS*, the key genes involved in JA synthesis, with both changing from asymmetrical to symmetrical. These findings indicate that grafting significantly changes the expression patterns of these two genes and may play an important role in grafting incompatibility.

The present study shows that *COI1* is the only gene in the JA signaling pathway that is affected by grafting, i.e., the expression patterns in the scions and rootstocks, were asymmetrical and was caused by wounding and later exhibit symmetrical expression after grafting. *COI1* plays an important role as a co-receptor with *JAZ* in the JA signal pathway [31]. In the damage treatment of *Aquilaria sinensis*, *AsCOI1* showed a high increase in expression at 1 h, about 20-fold higher, and the highest expression level at 6 h, about 33-fold higher [32]. In addition, the expression of *AaCOI1* in the leaves of *Artemisia annua* was significantly upregulated 6 h after wounding [33]. *SlCOI1* is essential for JA induced gene expression in tomatoes, and the JA signaling pathway with *SlCOI1* as the co-receptor plays a different role in controlling the expression levels of early and late response genes in wounding responses [34]. Therefore, the analysis of mutants may be essential to elucidate the function of *SlCOI1* in the successful healing of grafts.

The well characterized wounding response genes in mechanically damaged tomato leaves were divided into two categories based on temporal changes, i.e., early response and late response [35]. Early wound-response gene mRNA levels are upregulated 0.5 to 2 h, whereas those of late wound response genes increase from 4 to 24 h after injury [16, 36]. After mechanical injury in tomato leaves and seedling [37, 38], *SlLAPA* was activated at 2 h and reached the highest level at 8 h, i.e., *SlLAPA* is classified as a late response gene [16, 39]. However, in our results, *SlLAPA* showed an early response after grafting hypocotyl of tomato, and at the same time, its expression changed from asymmetrical in the separated part to symmetrical after grafting. This study demonstrated that the tomato wound response and grafting are a complex process.

The last gene, whose expression changed from asymmetrical before grafting to symmetrical after, was *SlJA2L*, which functions as a NAC transcription factors in JA mediated stomatal reopening [40]. *SlJA2L* is preferentially induced by mechanical wounding, and *SlMYC2*, *SlJA2L*, and *SlTD* expressions were induced by wounding and showed a temporal pattern, gradually increasing over time [16]. Chronologically, *SlMYC2* expression peaked at 0.5 h after injury, *SlJA2L* expression peaked at 1 h after injury, and *SlTD* expression peaked at 12 h after injury. The highest expression levels of *SlMYC2*, *SlJA2L* and *SlTD* increased by 10-, 100-, and 1,000-fold, respectively. These results suggest that there is a mechanism by which *SlMYC2*-mediated *SlTD* expression becomes a cascade response through *SlJA2L*, thus amplifying the damage signal. In our current study, by analyzing the expression patterns of these three genes within 12 h of grafting, we found that at the bottom of the separated part, *SlMYC2* expression peaked at 1 h, and both *SlJA2L* and *SlTD* peaked at 2 h, but *SlTD* peak expression was 100-fold higher than *SlJA2L*. At the top of the separated part, *SlMYC2* expression peaked at 0.5 h, *SlJA2L* at 2 h, and *SlTD* at 6 h. At this peak, the expression levels were 1.8-, 5.8-, and 580-fold higher, respectively. After grafting, *SlMYC2* expression peaked at 0.5 h in the rootstock, *SlJA2L* at 1 h, and *SlTD* at 2 h. In the scion, both *SlMYC2* and *SlJA2L* expression peaked at 1 h, and *SlTD* peaked at 6 h, with expressions levels 2.2-, 2.5-, and 230-fold higher respectively. In terms of temporal expression, grafting makes the cascade effect of these genes in rootstock more obvious, but reduces the amplification effect of expression level in the scions. These findings indicate that grafting not only alters the asymmetry of *SlJA2L* expression between scions and rootstocks, but also reduces the temporal and sequential *SlMYC2-SlJA2L-SlTD* expression and hierarchical effect.

In conclusion, our work proved that after the hypocotyl is cut off, the top and bottom parts of the hypocotyl show distinct asymmetric responses to wounding, while successful grafting can significantly reduce this asymmetric response. Through this work, some of JA-related genes that change in asymmetrical expression as induced by wounding between scion and rootstocks were identified. Additional studies that further examine these responses are necessary.

## Materials and methods

### Plant materials and growth conditions

Tomato (*Solanum lycopersicum*) cultivar Zhongshu No.4 seeds were surface sterilized with 70% ethanol for 5 s and 10 min in sterilization solution containing 35% commercial bleach solution (5.25% [w/v] sodium hypochlorite) and 0.1% Tween-20 (Bio-Rad, Shanghai, China). The seeds were placed in 1/2 strength MS medium containing 0.8% agarose. The germinated seedlings were grown for 5 to 7 d before grafting in growth chambers and maintained under 16 h of light at 25˚C and 8 h of dark at 18˚C and 60% relative humidity.

### Tomato hypocotyl grafting procedure

Tomato micrografting and grafting assays were performed according to previously published protocols [4] (For details see dx.doi.org/10.17504/protocols.io.bmhwk37e). The grafted tomato seedlings were placed in the growth chambers at a slightly inclined angle for 0, 0.5, 1, 2, 6, 12 h and inspected regularly. The tomato seedlings grafted at different time points were assessed under a microscope, the plants were gently moved with tweezers, and only those plants that were not separated from the junction were used as samples for successful grafting. When taking samples at different time points after grafting, the scion was carefully and forcibly removed from the rootstock with forceps at the joint site. Then, 5 mm of material from the grafting

interface were cut with a blade and used as sample of the scion and rootstock. For the sampling of the top and the bottom part of the material cut off separated without grafting at different time points after separated, the same material 5 mm from the fracture surface was prepared for grafting by cutting and the top and bottom parts of the sample were separated. Then, the material was placed into a corresponding labeled EP tube pre-cooled after weighing and again immediately weighed again. Finally, the sample tube was placed in liquid nitrogen, flash frozen, and stored at -80˚C until use.

## Quantification of JA and JA-Ile

The extraction and quantification of endogenous JA and JA-Ile were conducted according to the manufacturer's instructions (Wuhan Metware Biotechnology Co., Ltd., Wuhan, China). Around 1.0 g of each tomato sample was rapidly frozen in liquid nitrogen and homogenized into a powder using a mixer mill (MM 400, Retsch, Germany) for 1 min at 30 Hz. Using an LC-ESI-MS/MS system (HPLC, Shim-pack UFLC SHIMADZU CBM30A system, www.shimadzu.com.cn/; MS, Applied Biosystems 6500 Triple Quadrupole, www.appliedbiosystems.com.cn/) to analyze, the analytical conditions were as follows, HPLC: column, Waters ACQUITY UPLC HSS T3 C18 (1.8 µm, 2.1 mm*100 mm); solvent system, water (0.04% acetic acid): acetonitrile (0.04% acetic acid); gradient program, 90:10V/V at 0 min, 40:60 V/V at 5.0 min, 40:60 V/V at 7.0 min, 90:10 V/V at 7 min, 90:10 V/V at 10 min; flow rate, 0.35 mL/min; temperature, 40˚C; injection volume: 2 µL. The effluent was alternatively connected to an ESI-triple quadrupole-linear ion trap (Q TRAP)-MS. API 6500 Q TRAP LC/MS/MS System, equipped with an ESI Turbo Ion-Spray interface, operated in both positive and negative ion modes and controlled by Analyst 1.6 software (AB Sciex). The ESI source operation parameters were as follows: ion source, turbo spray; source temperature 500˚C; ion spray voltage (IS) 4500 V; curtain gas (CUR) were set at 35.0 psi; the collision gas (CAD) was medium. DP and CE for individual MRM transitions were done with further DP and CE optimization. A specific set of MRM transitions were monitored for each period according to the plant hormones eluted within this period. JA and JA-Ile contents were determined using the external standard method and expressed as lg/g fresh weight (FW) and ng/g FW, respectively. Three biological replications were performed.

## RNA extraction

Total RNA was extracted from Tomato hypocotyl samples using an RNAiso® plus reagent (TaKaRa) and reverse-transcribed into cDNAs with a RevertAid® First Strand cDNA Synthesis Kit (Thermo Scientific) according to the manufacturer's specifications. The yield of RNA was determined using a Nano-Drop 2000 spectrophotometer (Thermo Scientific, USA), and integrity was evaluated using agarose gel electrophoresis stained with ethidium bromide.

## Real-time quantitative RT-PCR

Quantification was performed with a two-step reaction process: reverse transcription (RT) and PCR. Each RT reaction has two steps. The first step consisted of the following: 0.5 µg RNA, 2 µL of 4×gDNA wiper Mix, add nuclease-free $H_2O$ to 8 µL. Reactions were performed in a GeneAmp® PCR System 9700 (Applied Biosystems, USA) for 2 min at 42˚C. The second step involved add 2 µL of 5 × HiScript II Q RT SuperMix IIa. Reactions were performed in a GeneAmp® PCR System 9700 (Applied Biosystems, USA) for 15 min at 50˚C and 5 s at 85˚C. A 10-µL RT reaction mix was then diluted × 10 in nuclease-free water and held at -20˚C. Real-time PCR was performed using LightCycler® 480 II Real-time PCR Instrument (Roche, Swiss) with 10 µL of the PCR reaction mixture that included 1 µL of cDNA, 5 µL of 2×ChamQ

SYBR qPCR Master Mix, 0.2 μL of forward primer, 0.2 μL of reverse primer and 3.6 μL of nuclease-free water. Reactions were incubated in a 384-well optical plate (Roche, Swiss) at 95˚C for 30 s, followed by 40 cycles of 95˚C for 10 s, 60˚C for 30 s. Each sample was run in triplicate for analysis. At the end of the PCR cycles, melting curve analysis was performed to validate the specificity in generating the expected PCR products. The primer sequences were designed and synthesized by Generay Biotech (Generay, PRC) based on the mRNA sequences obtained from the NCBI database: (S3 Table).

The mRNA expression levels were normalized to the *TUA* gene of tomato (*S. lycopersicum* elongation factor *TUA*, NCBI RefSeq: XM_010320721.3) and were calculated using the $2^{-\Delta\Delta Ct}$ method ($\Delta Ct = Ct_{Target} - Ct_{TUA}$; $\Delta\Delta Ct = \Delta Ct_{Experimental} - \Delta Ct_{Control}$) [41].

## Gene expression correlation analysis

The correlation and the test *P* values between the upper and lower parts of the plant and gene expression levels before and after grafting were calculated using SPSS software, and correlation coefficient $r^2$ was calculated using Graph Prism 5 software.

## Qualitative and quantitative analysis of plant hormones

After separation and detection using UPLC (Ultra Performance Liquid Chromatog-raphy, UPLC) (Shim-pack UFLC SHIMADZU CBM30A) and MS/MS (Tandem mass spectrometry, MS/MS) (Applied Biosystems 6500 Quadrupole Trap), the software Analyst 1.6.1 was used to process mass spectrometry data. According to the information of JA and JA-Ile retention time and peak type, the mass spectrum peaks detected by each hormone in different samples were corrected to ensure the accuracy of qualitative and quantitative. The JA and JA-Ile of all samples were analyzed, with the peak area of each chromatographic peak representing the relative content of the corresponding hormone, and the qualitative and quantitative analysis results of the JA and JA-Ile of all samples were obtained.

## Statistical analysis

The data of hormone contents and genes relative expression level were statistically analyzed by the Excel software (version Microsoft Office 365, London, UK). Student's *t*-test was used on the data sets and tested for significant ($P \leq 0.05$ and $P \leq 0.01$) differences analysis.

The JA-related genes expression patterns correlation r was statistically analyzed by the Excel software (Microsoft Office 365, London, UK). Pearson correlation coefficient, $0.8 \leq |r| \leq 1$, highly correlated; $0.5 \leq |r| < 0.8$, moderately correlated; $0.3 \leq |r| < 0.5$, weakly correlated, and $|r| < 0.3$, negligible correlated. *P* value was statistically analyzed by the Excel software (Microsoft Office 365, London, UK). The significant was $P \leq 0.05$ and $P \leq 0.01$. The determining coefficient $r^2$ were statistically analyzed by the Graphpad prism (version GraphPad Software Inc., USA).

## Supporting information

**S1 Table. Dynamic content data of tomato jasmonate (JA) and jasmonate isoleucine conjugate (JA-Ile) within 12 hours after grafting.**
(PDF)

**S2 Table. Data of dynamic expression of 30 JA-related genes within 12 hours after grafting.**
(PDF)

**S3 Table. The primers of 30 JA-related genes used in this study.**
(PDF)

**S1 Fig. Dynamic changes in the expression of JA synthesis-related genes within 12 h of grafting.**
(TIF)

**S2 Fig. Dynamic changes in expression of JA signaling genes within 12 h of grafting.**
(TIF)

**S3 Fig. Dynamic changes in expression of defense genes within 12 h of grafting.**
(TIF)

**S4 Fig. Dynamic changes in expression of JA-regulated genes within 12 h of grafting.**
(TIF)

## Author Contributions

**Conceptualization:** Jiaqi Wang, Dongliang Li, Ni Chen, Jingjing Chen, Changjun Mu, Kuide Yin, Yuke He, Heng Liu.

**Data curation:** Jiaqi Wang, Dongliang Li, Ni Chen, Jingjing Chen, Changjun Mu, Kuide Yin, Heng Liu.

**Formal analysis:** Jiaqi Wang, Dongliang Li, Ni Chen, Jingjing Chen, Changjun Mu, Kuide Yin, Heng Liu.

**Funding acquisition:** Heng Liu.

**Investigation:** Jiaqi Wang, Ni Chen, Changjun Mu, Kuide Yin, Yuke He, Heng Liu.

**Methodology:** Jiaqi Wang, Dongliang Li, Ni Chen, Jingjing Chen, Heng Liu.

**Project administration:** Jiaqi Wang, Kuide Yin, Yuke He, Heng Liu.

**Resources:** Jiaqi Wang, Heng Liu.

**Software:** Jiaqi Wang, Dongliang Li, Ni Chen, Heng Liu.

**Supervision:** Kuide Yin, Yuke He, Heng Liu.

**Validation:** Jiaqi Wang, Dongliang Li, Ni Chen, Jingjing Chen, Changjun Mu, Kuide Yin, Heng Liu.

**Visualization:** Jiaqi Wang, Dongliang Li, Ni Chen, Jingjing Chen, Changjun Mu, Heng Liu.

**Writing – original draft:** Jiaqi Wang, Dongliang Li, Ni Chen, Jingjing Chen, Changjun Mu, Kuide Yin, Yuke He, Heng Liu.

**Writing – review & editing:** Jiaqi Wang, Ni Chen, Jingjing Chen, Changjun Mu, Kuide Yin, Yuke He, Heng Liu.

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
