## [Decision Letter · Decision Letter 0]

28 Aug 2020

PONE-D-20-22029

Plant grafting relieves asymmetry of jasmonic acid response induced by wounding between scion and rootstock in tomato hypocotyl

PLOS ONE

Dear Dr. Liu,

Thank you for submitting your manuscript to PLOS ONE. After careful consideration, we feel that it has merit but does not fully meet PLOS ONE’s publication criteria as it currently stands. Therefore, we invite you to submit a revised version of the manuscript that addresses the points raised during the review process.

Please improve the manuscript according to the reviewers' comments.

We look forward to receiving your revised manuscript.

Kind regards,

Yonggen Lou

Academic Editor

PLOS ONE

Journal Requirements:

Reviewers' comments:

Reviewer's Responses to Questions

**Comments to the Author**

1. Is the manuscript technically sound, and do the data support the conclusions?

Reviewer #1: Yes

Reviewer #2: Partly

2. Has the statistical analysis been performed appropriately and rigorously? 

Reviewer #1: I Don't Know

Reviewer #2: Yes

3. Have the authors made all data underlying the findings in their manuscript fully available?

Reviewer #1: Yes

Reviewer #2: Yes

4. Is the manuscript presented in an intelligible fashion and written in standard English?

Reviewer #1: Yes

Reviewer #2: No

5. Review Comments to the Author

Reviewer #1: In this MS, Wang et al. compared the levels of JA and JA-Ile and the transcript accumulation of JA-related genes in scions and rootstocks of tomato seedlings and in the non-grafted (separated) top and bottom parts of tomato seedlings. They found that grafted top and bottom parts had more similar profiles of JA/JA-Ile and JA-related transcripts than did the separated top and bottom parts. This study provides new insight into the mechanism of grafting, which is important in agriculture.

I have a few minor comments.

1. The readability of figures should be improved. They are too small and too crowded. Instead of line charts, I would recommend using bar charts, which are more reasonable to use here.

2. line 156, change “6 JA synthesis-related genes” to “six JA synthesis-related genes”. When talking about quantities less than 10, it is better not to use numerals.

3. It is interesting that the JA-Ile content showed a "W"-shaped response in 0-6 h. As described in Figure 1B, the JA levels both increased 3 or 4 times in the bottom and top part, but in Figure 1C, the JA-Ile level showed a strong decrease after wounding, although it returned to the basal level after grafting 6 h. Normally, the JA and JA-Ile content are consistently induced by the wounding (as in line 371, after wounding, the content of JA-Ile has peak expression at 1h in tomato leaves). A discussion is needed for this result.

4. Lines 387-388, “These findings indicate that the restoration of the priming of JA-Ile plays an important role in the success of grafting”. The presented data do not really support this claim.

5. Statistical method should be indicated in the figure legends and MM.

Reviewer #2: Plant grafting is a sequential wounding healing process. The paper showed dynamic profile of JA and JA-Ile concentration in scion and rootstock at different time point after grafting, transcriptional expression of a series of JA-related genes, and found that grafting alleviates the asymmetrical JA and defense and so on. It is so interesting work, but the manuscript should be major revised including the writing before being accepted. Some questions or suggestion:

1. From Fig.2 to Fig.5, no sign is marked to certain gene so I could not distinguish the transcriptional expression levels of genes and understand the meaning of paper well.

2. As the scion could acquire water and others from rootsock, maybe need to set other treatment group rather than the separated top sample of scion without other treatment in order to further consolidate the conclusion.

3. In addition, the list of references is not in the same style.

4. All legends of figures need to be organized.

6. PLOS authors have the option to publish the peer review history of their article (what does this mean?). If published, this will include your full peer review and any attached files.

Reviewer #1: No

Reviewer #2: No

---

## [Author Response · Author response to Decision Letter 0]

20 Sep 2020

Dear Prof. Lou and reviewers,

We would like to thank you for the time and effort that you have spent reviewing our manuscript. We are pleased to note that you have found our research on the mechanism of grafting is interesting and important and may provide some of new insight for the further research in this field. We are also very pleased to thank you for the comments pointed out, which will help us to improve the quality of our work.

In view of your motivated comments, we have reconsidered deeply and sorted out all the suggestions mentioned. In particular, this revised manuscript has significantly been improved. Meanwhile, our itemized responses to your questions, comments and suggestions are mainly as following (repeated below in italics for your convenience):

Reviewer 1

1. The readability of figures should be improved. They are too small and too crowded. Instead of line charts, I would recommend using bar charts, which are more reasonable to use here.

Reply：Thank you for the reasonable comments. The line charts have been modified to the bar charts in accordance with the comment. Meanwhile, all the figures have been uploaded and checked to the Preflight Analysis and Conversion Engine (PACE) digital diagnostic tool, to make sure all the figures to meet PLOS requirements.

See “Fig 1-5” for details please.

2. line 156, change “6 JA synthesis-related genes” to “six JA synthesis-related genes”. When talking about quantities less than 10, it is better not to use numerals.

Reply: We have changed the numbers less than 10 to English as comment. 

The modified position in the “Manuscript” file is lines 155, 195 and 226, and in the “Revised

Manuscript with Track Changes” file is lines 159, 204 and 241.

3. It is interesting that the JA-Ile content showed a "W"-shaped response in 0-6 h. As described in Figure 1B, the JA levels both increased 3 or 4 times in the bottom and top part, but in Figure 1C, the JA-Ile level showed a strong decrease after wounding, although it returned to the basal level after grafting 6 h. Normally, the JA and JA-Ile content are consistently induced by the wounding (as in line 371, after wounding, the content of JA-Ile has peak expression at 1h in tomato leaves). A discussion is needed for this result.

Reply：This is a very good suggestion about our result of JA-Ile. According to your comment, we have discussed it more deeply, the details are as follows:

Original description before modification：Unlike the time response curve of JA, JA-Ile appeared at 2 h after grafting; grafting obviously restored the level of JA Ile in the scions and rootstocks. These findings indicate that the restoration of the priming of JA-Ile plays an important role in the success of grafting.

Description after revision：Unlike the time response curve of JA, our most interesting results was that JA-Ile content showed a “W”-shaped response in 0-6 h. Meanwhile, grafting obviously initiated to restore the level of JA Ile in the scions and rootstocks from 2 h after grafting. In leaves, roots and shoots, generally, the JA and JA-Ile content are consistently induced by the wounding [21-23], and no JA-Ile reduction was reported within half an hour after injury in these tissues yet. We do not know whether the W-shaped JA-Ile response induced by wounding is specific to the hypocotyl of tomatoes. Therefore, further researches are necessary to confirm the ubiquity of this result in stem grafting of tomatoes and other species including Arabidopsis thaliana. Moreover, further analysis on various JA synthesis mutants will certainly contribute to our understanding of this issue.

The modified position in the “ Manuscript ” file is lines 382-392 ， and in the “ Revised Manuscript with Track Changes ” file is lines 405-415.

4. Lines 387-388, “These findings indicate that the restoration of the priming of JA-Ile plays an important role in the success of grafting”. The presented data do not really support this claim.

Reply: This comment is correct and valuable. We still need more research results to support the role of JA-Ile in the processes of grafting development. Therefore, in the reply to the third

question and this question, we are trying our best to explained and discussed this question

together. Detailed as follows.

Original description before modification：These findings indicate that the restoration of the

priming of JA-Ile plays an important role in the success of grafting.

Description after supplement：Unlike the time response curve of JA, our most interesting results was that JA-Ile content showed a “W”-shaped response in 0-6 h. Meanwhile, grafting obviously initiated to restore the level of JA Ile in the scions and rootstocks from 2 h after grafting. In leaves, roots and shoots, generally, the JA and JA-Ile content are consistently induced by the wounding [21-23], and no JA-Ile reduction was reported within half an hour after injury in these tissues yet. We do not know whether the W-shaped JA-Ile response induced by wounding is specific to the hypocotyl of tomatoes. Therefore, we need to further study to confirm the ubiquity of this result in stem grafting of tomatoes and other species including Arabidopsis thaliana. Moreover, further analysis on various JA synthesis mutants will certainly contribute to our understanding of this issue.

The modified position in the “ Manuscript ” file is lines 382-392，and in the “Revised Manuscript with Track Changes” file is lines 405-415.

5. Statistical method should be indicated in the figure legends and MM.

Reply: The statistical method has been supplemented in the figure legends and MM according to the comments.

The modified position in the “Manuscript” file is lines150-154, 191-194, 222-225, 267-271, 302-306, 584-585, 598-609, and in the “Revised Manuscript with Track Changes” file is lines150-154, 195-198, 232-235, 282-285, 321-324, 611-612, 629-643.

Reviewer 2

1. From Fig.2 to Fig.5, no sign is marked to certain gene so I could not distinguish the transcriptional

expression levels of genes and understand the meaning of paper well.

Reply: The genes in Fig.2 to Fig.5 have been marked as suggestion. See “Fig 1-5” file for details please.

2. As the scion could acquire water and others from rootsock, maybe need to set other treatment group rather than the separated top sample of scion without other treatment in order to further consolidate the conclusion.

Reply: This is a very good and constructive opinion. The scion obtained water and inorganic salt from the scion, and the scion obtained photosynthetic product from the scion. The supply of water and inorganic salts to scions and the supply of photosynthetic products to rootstock can be restored only after vascular system reconnection. Within 12 hours after grafting, the wounding response is predominant and the vascular system has not yet been reconnected. For Arabidopsis the reconnection starts about 48 to 72 hours, and for tomatoes it starts about 72 hours later. Therefore, this study focuses on the 12 hours after grafting. After the relationship between early wounding response and grafting is clarified, then next step we will focus on the processes of rebuilding the vascular system and use various treatments to study the relationship between water, inorganic salts and photochemical products and grafting.

3. In addition, the list of references is not in the same style.

Reply: The overall format of the references has been revised in accordance with the comment and PLOS ONE style requirements. The main modification was to delete the issue numbers, shorten the interval between the serial numbers and the reference, and modify the page number format, font size and line spacing.

The modified position in the “Manuscript” file is lines 631-748, and in the “Revised Manuscript

with Track Changes” file is lines 666-783.

4. All legends of figures need to be organized.

Reply: The legends of the figures and tables have been reorganized according to the comments and PLOS ONE style requirements.

The specific modified position in the “Manuscript” file is lines 150-154, 191-194, 222-225, 267-271, 302-306, 330-333, 335, 337-340, 342, 344-347, 349, 351-354, 356, and in the “Revised Manuscript with Track Changes” file is lines 150-154, 195-198, 232-235, 282-285, 321-324, 353-356, 358, 360-363, 365, 367-370, 372, 374-377, 379.

Other modifications: We have additionally made reasonable modifications to the author; E-mail and the text title parts of our manuscript.

The modified position in the “Manuscript” file is lines 6-7, 26-27, 307, and in the “Revised

Manuscript with Track Changes” file is lines 6-7, 26-27, 330.

According to the comment, we have deposited our laboratory protocol about tomato hypocotyl grafting in protocol.io, and DOI has been cited in our manuscript already. Please see MM in the “Manuscript” file and the “ Revised Manuscript with Track Changes ” file. The modified position in the “Manuscript” file is lines 517, and in the “Revised Manuscript with Track Changes” file is lines 544.

Thank you very much for your time and consideration.

Sincerely yours,

Heng Liu (Corresponding Author)

---

## [Decision Letter · Decision Letter 1]

13 Oct 2020

Plant grafting relieves asymmetry of jasmonic acid response induced by wounding between scion and rootstock in tomato hypocotyl

PONE-D-20-22029R1

Dear Dr. Liu,

We’re pleased to inform you that your manuscript has been judged scientifically suitable for publication and will be formally accepted for publication once it meets all outstanding technical requirements.

Kind regards,

Yonggen Lou

Academic Editor

PLOS ONE

Additional Editor Comments (optional):

Reviewers' comments:

Reviewer's Responses to Questions

**Comments to the Author**

1. If the authors have adequately addressed your comments raised in a previous round of review and you feel that this manuscript is now acceptable for publication, you may indicate that here to bypass the “Comments to the Author” section, enter your conflict of interest statement in the “Confidential to Editor” section, and submit your "Accept" recommendation.

Reviewer #1: All comments have been addressed

Reviewer #2: All comments have been addressed

2. Is the manuscript technically sound, and do the data support the conclusions?

Reviewer #1: Yes

Reviewer #2: Partly

3. Has the statistical analysis been performed appropriately and rigorously? 

Reviewer #1: Yes

Reviewer #2: No

4. Have the authors made all data underlying the findings in their manuscript fully available?

Reviewer #1: Yes

Reviewer #2: No

5. Is the manuscript presented in an intelligible fashion and written in standard English?

Reviewer #1: Yes

Reviewer #2: Yes

6. Review Comments to the Author

Reviewer #1: (No Response)

Reviewer #2: The paper described a so interesting result that asymmetry of jasmonic acid between scion and rootstock after grafting. However, the conclusion could not be supported or explained clearly and logically.

Major comments:

1.How asymmetry of jasmonic acid between scion and rootstock after grafting were not well defined and described in results, 'asymmetry of jasmonic acid' means different levels between scion and rootstock after grafting, or between grafting materials and seperated.

2. If 'asymmetry of jasmonic acid' means different levels between scion and rootstock after grafting. Maybe in Figure 1, only the data of grafted top and bottom were compared and showed will be more clear, or try to compare JA levels of four treatments rather than in two independent figure in figure 1B and 1C.

3. For figure 2-5, the expression levels of important genes were selected to show, others could be moved to supplemental materials.

4.in materals and methods, 'qualitative and quantitative analysis of plant hormones' should be described in more details, such as using what instrument HPLC or GC-MS, which hormones were detected?

Minor comments:

1. In line 140, 'after 2 h' is not accurate.

2. All the figure legends in text is not suitable. In general, figure legends were placed after reference.

3. the letter 'r' should be italic like 'P' in Line 334-335, Line 341-342 and so on.

4.Line 384 check the 'JA Ile'

5.Line 390 'Arabidopsis thaliana' is not italic.

6. The layout of references is not the same. For reference No.7,8 ..., the all first letter of title is capital, others only the first letter of the first word of title is capital.

7. PLOS authors have the option to publish the peer review history of their article (what does this mean?). If published, this will include your full peer review and any attached files.

Reviewer #1: No

Reviewer #2: No

---

## [Editor Report · Acceptance letter]

13 Nov 2020

PONE-D-20-22029R1 

Plant grafting relieves asymmetry of jasmonic acid response induced by wounding between scion and rootstock in tomato hypocotyl 

Dear Dr. Liu:

I'm pleased to inform you that your manuscript has been deemed suitable for publication in PLOS ONE. Congratulations! Your manuscript is now with our production department. 

Kind regards, 

on behalf of

Dr. Yonggen Lou 

Academic Editor

PLOS ONE